# Shh from mossy cells contributes to preventing NSC pool depletion after seizure-induced neurogenesis and in aging

Hirofumi Noguchi[1], Jessica Chelsea Arela[1], Thomas Ngo[1], Laura Cocas[1,2], Samuel Pleasure[1,3]*

[1]Department of Neurology, University of California, San Francisco, San Francisco, United States; [2]Santa Clara University, Biology Department, Neuroscience Program, Santa Clara, United States; [3]Programs in Neuroscience and Developmental & Stem Cell Biology, Eli and Edythe Broad Center of Regeneration Medicine and Stem Cell Research, University of California San Francisco, San Francisco, United States

*For correspondence:
Samuel.Pleasure@ucsf.edu

**Abstract** Epileptic seizures induce aberrant neurogenesis from resident neural stem cells (NSCs) in the dentate gyrus of the adult mouse hippocampus, which has been implicated in depletion of the NSC pool and impairment of hippocampal function. However, the mechanisms regulating neurogenesis after seizures remain unknown. Here, we demonstrate that Sonic hedgehog (Shh) from mossy cells is a major source of Shh signaling activity after seizures, by which mossy cells contribute to seizure-induced neurogenesis and maintenance of the NSC pool. Deletion of *Shh* from mossy cells attenuates seizure-induced neurogenesis. Moreover, in the absence of Shh from mossy cells, NSCs pool are prematurely depleted after seizure-induced proliferation, and NSCs have impaired self-renewal. Likewise, lack of Shh from mossy cells accelerates age-related decline of the NSC pool with accompanying reduction of self-renewal of NSCs outside the context of pathology such as seizures. Together, our findings indicate that Shh from mossy cells is critical to maintain NSCs and to prevent exhaustion from excessive consumption in aging and after seizures.

## eLife assessment

This study uses specific and robust genetic approaches to assess mechanisms of kainic acid-induced neurogenesis. This is a **fundamental** study that bridges several complementary methods and is a **convincing** use of existing approaches to explore roles for sonic hedgehog in activity-dependent and aging-associated hippocampal neurogenesis.

## Introduction

In the dentate gyrus (DG) of the adult hippocampus, neural stem cells (NSCs) are preserved beyond development in a special niche where newborn neurons are generated and integrated into the hippocampal circuit throughout life (*Berg et al., 2019*; *Deng et al., 2010*; *Imayoshi et al., 2008*; *Ming and Song, 2011*). NSCs in the adult DG are maintained in a quiescent state, thus adult neurogenesis begins by activating NSC proliferation, followed by differentiation and maturation of newborn neurons (*Encinas et al., 2011*; *Lugert et al., 2010*; *Urbán et al., 2019*). NSC proliferation and subsequent differentiation in the adult DG are influenced by neuronal activity and the processing of multiple external stimuli and have been shown to be dysregulated after brain insults and in disease states

(*Kempermann, 2019*; *Lugert et al., 2010*; *Moreno-Jiménez et al., 2019*; *van Praag et al., 1999a*; *Sierra et al., 2015*; *Tobin et al., 2019*).

Epileptic seizures are known to strongly induce adult neurogenesis (*Jessberger et al., 2007*; *Jessberger and Parent, 2015*; *Parent et al., 1997*). Normally, increased neurogenesis has been shown to contribute to hippocampus-dependent learning and memory (*Deng et al., 2010*; *Ming and Song, 2011*). However, seizure-induced newborn neurons have morphological abnormalities, and many ectopically migrate into the hilus of the DG (*Parent et al., 1997*) contributing to formation of aberrant neuronal circuits in the hippocampus. These ectopic neurons are implicated in cognitive impairment as well as in the development of repeated spontaneous seizures (*Cho et al., 2015*; *Lybrand et al., 2021*). Furthermore, previous studies showed that once NSC proliferation is activated, NSCs are consumed after a series of cell divisions for neuronal production, resulting in the reduction of neurogenesis and the decline of the NSC pool with age (*Encinas et al., 2011*; *Harris et al., 2021*; *Lugert et al., 2010*). Excessive neuronal production induced by neuronal hyperactivity has been linked to accelerated consumption of NSCs and leads to depletion of the NSC pool (*Fu et al., 2019*; *Sierra et al., 2015*). Despite these studies, little is known about the mechanisms by which seizures induce neurogenesis and how NSCs are regulated in pathogenic conditions to result in NSC pool depletion.

NSCs are regulated by numerous signaling molecules from other niche cells (*Bonafina et al., 2020*; *Morales and Mira, 2019*). Dysregulation of the NSC niche in pathological conditions has been suggested to contribute to aberrant NSC behavior (*Li and Guo, 2021*; *Salta et al., 2023*). There are several different types of neurons in close proximity to NSCs in the DG. A recent series of studies demonstrated that hippocampal neuronal networks between these niche neurons regulate neuronal activity-induced neurogenesis (*Asrican et al., 2020*; *Bao et al., 2017*; *Song et al., 2016*; *Song et al., 2013*; *Song et al., 2012*; *Yeh et al., 2018*). Mossy cells are excitatory neurons in the dentate hilus that provide glutamatergic inputs to newborn granule neurons and GABAergic interneurons in the DG (*Chancey et al., 2014*; *Scharfman, 2016*). Mossy cells are among the most sensitive neurons to afferent excitation in the hippocampus and are vulnerable to continuous seizures (*Scharfman, 1991*; *Scharfman, 2016*; *Sloviter et al., 2003*). Mossy cell activity modulates NSC proliferation depending on the intensity of activation: only highly activated mossy cells increase NSC proliferation (*Yeh et al., 2018*). These studies suggest a potential connection between seizure-induced mossy cell hyperexcitation and the aberrant neurogenesis seen after seizures.

We previously found that Sonic hedgehog (Shh), a morphogen inducing cell proliferation, is expressed by mossy cells (*Li et al., 2013*). Dysregulation of Shh signaling activity has been reported in epileptic seizures where increased Shh signaling and Shh expression were observed in the hippocampus of temporal lobe epilepsy patients and epileptic rodent models (*Fang et al., 2011*; *Pitter et al., 2014*; *Zhang et al., 2017*). Manipulation of Shh receptors in NSCs has shown that increased Shh signaling in NSCs drives NSC proliferation and neurogenesis (*Antonelli et al., 2018*; *Choe and Pleasure, 2013*; *Daynac et al., 2016*). These findings may indicate contributions of Shh in seizure-induced neurogenesis. However, the source and regulation of Shh that regulates neurogenesis, especially during seizure activity, remains undefined. Direct evidence connecting Shh signaling and neural activity with seizure-induced aberrant neurogenesis has not been examined.

In this study, we identify mossy cells as a major source of Shh during epileptic seizures and find that activation of Shh signaling promotes neurogenesis following seizures. Seizure-induced neurogenesis is attenuated by selective deletion of *Shh* in mossy cells. Our data also show that Shh from mossy cells is needed to preserve the NSC pool after seizure-induced neurogenesis. Interestingly, even though seizure-induced neurogenesis is reduced in the absence of Shh from mossy cells, deletion of *Shh* also results in premature depletion of the NSC pool after seizures. We find that Shh from mossy cells is important for NSCs to return to the stem cell state after seizure-induced proliferation. Furthermore, our data demonstrate that *Shh* deletion in mossy cells results in premature decline of NSC pool with age even without seizures, and that NSCs are less able to return to the stem cell state after proliferation without mossy cell supplied Shh. Together, our results indicate that Shh derived from mossy cells increases proliferation and self-renewal of NSCs during seizure-induced neurogenesis, thereby ensuring that the NSC pool is sustained.

# Results

## Shh signaling is activated by seizures and contributes to seizure-induced aberrant neurogenesis

To understand the role of Shh signaling in seizure-induced aberrant neurogenesis, we first investigated whether Shh signaling is activated upon induction of seizure activity. Expression of *Gli1*, a downstream transcription factor of Shh signaling, is transcriptionally induced upon activation of Shh signaling (*Bai et al., 2002*; *Lee et al., 1997*). In the subgranular zone (SGZ) of the DG, *Gli1* expression and activation of Shh signaling are seen exclusively in NSCs (*Ahn and Joyner, 2005*; *Bottes et al., 2021*). To confirm this, using *Gli1^{CreER}* mice crossed with Cre-dependent red fluorescent protein (tdTomato) reporter mice Rosa^{AI14} (*Ahn and Joyner, 2004*; *Madisen et al., 2010*), we labeled *Gli1* expressing cells in the DG with tdTomato. One day after a 3-day tamoxifen treatment, we observed that about 90% of tdTomato-labeled cells in the SGZ were Sox2+ S100B− cells (*Figure 1—figure supplement 1*), comprising Sox2+ GFAP+ radial glial cell population. This confirms that NSCs are Shh responsive and that *Gli1* expression in the SGZ is essentially restricted to NSCs. Thus, we used *Gli1^{nLacZ/+}* mice, which carry the nuclear LacZ (nLacZ) transgene driven by the promoter of *Gli1* (*Ahn and Joyner, 2005*; *Bai et al., 2002*; *Ihrie et al., 2011*) as a sensitive readout for the activation of Shh signaling in NSCs and examined the number of Gli1-nLacZ+ cells, which represent Shh-responding cells, in SGZ after seizure induction. We induced seizures with consecutive intraperitoneal injection of low-dose kainic acid (KA) in *Gli1^{nLacZ/+}* mice and found that Gli1-nLacZ+ Shh-responding cells in the SGZ are significantly increased 3 days post-seizure induction (*Figure 1A, B*). These data show that Shh signaling activity is increased in NSCs after KA-induced seizures.

To investigate whether Shh contributes to seizure-induced aberrant neurogenesis, we analyzed Shh knock-in heterozygote (*Shh^{EGFP-Cre/+}*, from here on referred as *Shh^{+/−}*) mice (*Harfe et al., 2004*). We found that Gli1-nLacZ+ Shh-responding cells in the SGZ of *Shh^{+/−}; Gli1^{nLacZ/+}* mice were significantly reduced (*Figure 1C, D*), indicating that the level of Shh signaling in NSCs is decreased in these mice. We also observed that Gli1-nLacZ+ cells failed to increase after seizure induction in these mice, indicating that Shh signaling activity is not upregulated with seizure activity in *Shh^{+/−}* mice. We then investigated whether seizures induced aberrant neurogenesis in *Shh^{+/−}* mice. After induction of seizures by KA, we injected 5-bromo-2′-deoxyuridine (BrdU) for 5 days to label newborn neurons and analyzed the number of BrdU-labeled DCX+ newborn neurons at 3 days after the last BrdU injection (*Figure 1E*). In WT mice, the number of BrdU+ DCX+ newborn neurons dramatically increased after seizure induction. However, this was significantly reduced in *Shh^{+/−}* mice by comparison (*Figure 1F–H*). Ectopic migration of newborn neurons into the hilus is known to occur in post-seizure neurogenesis (*Lybrand et al., 2021*; *Parent et al., 1997*). *Shh^{+/−}* mice have a reduction in ectopic hilar dentate neurons (*Figure 1I*). These results demonstrate that Shh contributes to seizure-induced aberrant neurogenesis.

## *Shh* expression is induced in hilar mossy cells in the DG by seizure induction

To further understand the role of Shh in seizure-induced aberrant neurogenesis, we sought to identify the source of Shh ligands in the DG, particularly after seizures. *Shh^{EGFP-Cre/+}* mice were crossed with Rosa^{AI14} reporter mice (from here on referred as *Shh^{Cre/+}*;Rosa^{AI14}) and seizures were induced. One week after seizure induction, we found that there was specific tdTomato expression in the dentate hilus of *Shh^{Cre/+}*;Rosa^{AI14} mice both with and without seizure induction (*Figure 2A, B*). Since inhibitory interneurons and mossy cells are both prominent dentate hilar cell types, we explored which of these are included in the tdTomato+ cells. Co-immunostaining with the cell-type markers showed that the majority of tdTomato+ cells were GluR2/3+ mossy cells but with a much smaller number of Parvalbumin+ interneurons expressing tdTomato (*Figure 2C, D*). Furthermore, the cell types reflected in the population of tdTomato+ cells did not change with seizure induction, thus indicating that in the hilus, seizures did not lead to ectopic expression of Shh in cells that normally do not express the ligand (*Figure 2D*). Interestingly, we found that tdTomato+ cells were dominantly observed in the dorsal GluR2/3+ mossy cells but not in the ventral hippocampus (*Figure 2—figure supplement 1*). These data suggest that dorsal mossy cells are the major source of Shh in DG in both the naive and post-seizure hippocampus. We also observed increased numbers of Shh-responding NSCs of the SGZ upon seizure induction (*Figure 1A, B*). Therefore, we next tested the possibility that the

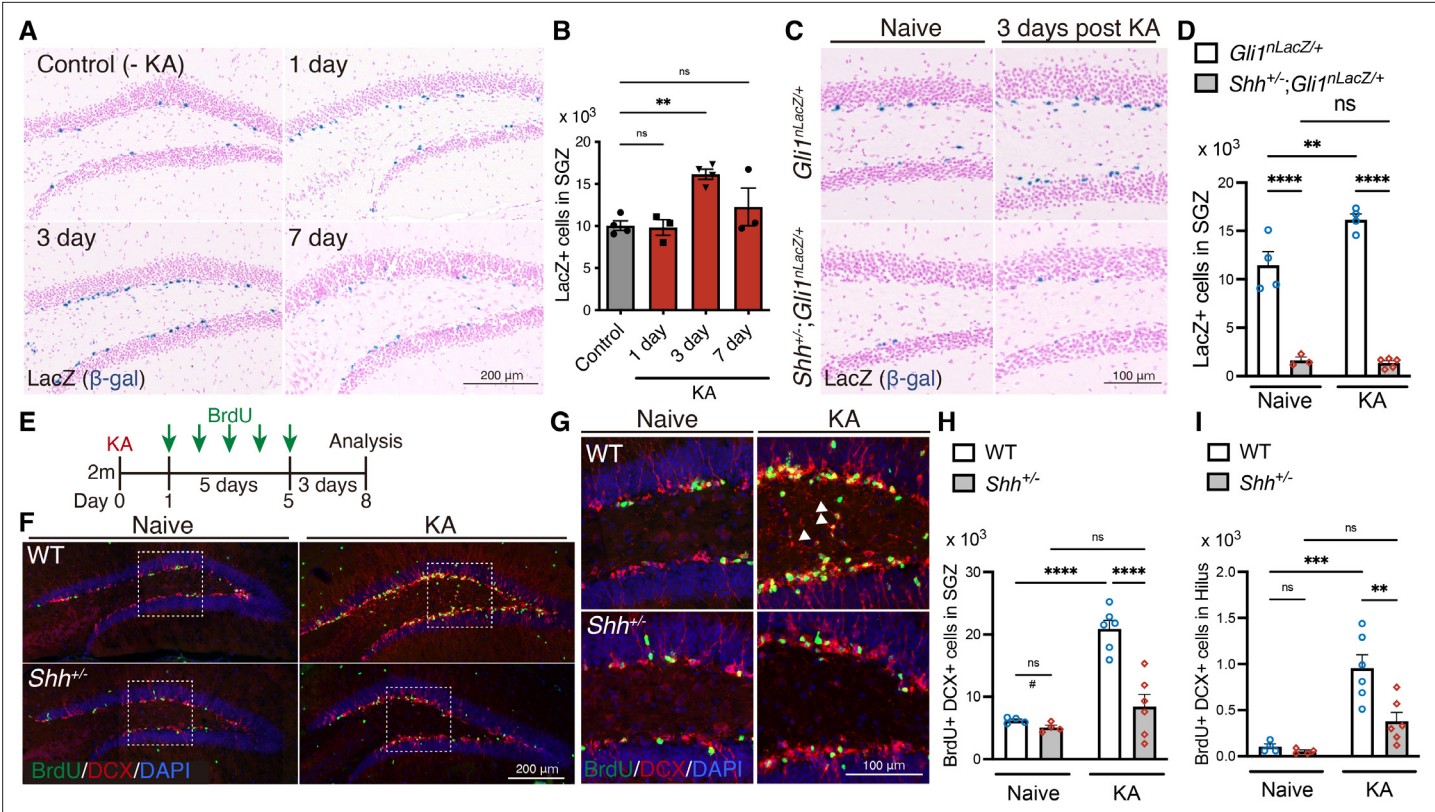

**Figure 1.** Sonic hedgehog (Shh) signaling activity is increased upon seizure induction. (**A**) Representative Gli1-nLacZ staining images of the SGZ of *Gli1^nLacZ/+^* mice 1, 3, or 7 days after kainic acid (KA)-induced seizure. Control mice did not receive KA treatment. (**B**) Quantification of Gli1-LacZ+ cells in SGZ. Values represent mean ± standard error of the mean (SEM); ns: p > 0.05, **p < 0.01. One-way analysis of variance (ANOVA) with Tukey's multiple comparison test (control: n = 4, 1 day: n = 3, 3 days: n = 4, 7 days: n = 3 mice). (**C**) Representative Gli1-LacZ staining from SGZ of *Gli1^nLacZ/+^* and *Shh^+/–^;Gli1^nLacZ/+^* mice 3 days after KA-induced seizure. (**D**) Quantification of Gli1-nLacZ+ cells in SGZ without (Naive) and with seizure induction (KA). Values represent mean ± SEM; ns: p > 0.05, **p < 0.01, ****p < 0.0001. Two-way ANOVA with Tukey's multiple comparison test (naive: *Gli1^nLacZ/+^* n = 4, *Shh^+/–^;Gli1^nLacZ/+^* n = 3, KA: *Gli1^nLacZ/+^* n = 4, *Shh^+/–^;Gli1^nLacZ/+^* n = 5 mice). (**E**) Experimental scheme of analyzing neurogenesis by 5-bromo-2'-deoxyuridine (BrdU) pulse labeling. Seizures were induced in 2-month-old (2 m) mice by KA injection. One day after KA-induced seizures, the mice received BrdU for 5 days and then were analyzed 3 days after the last BrdU injection. (**F**) Representative immunofluorescence images of newborn neurons labeled with BrdU (green), DCX(doublecortin) (red), and 4',6-diamidino-2-phenylindole, dihydrochloride (DAPI; blue) in the SGZ of wild-type (WT) and *Shh^+/–^* mice after seizure induction. (**G**) Higher magnification images from inset of panel (**F**), representing ectopic neurons in the hilus, which are indicated by white arrowheads. Quantification of newborn neurons (**H**) and ectopic neurons (**I**) produced after seizure induction in the SGZ of WT and *Shh^+/–^* mice. Values represent mean ± SEM; ns: p > 0.05, **p < 0.01, ***p < 0.001, ****p < 0.0001. Two-way ANOVA with Tukey's multiple comparison test (naive: WT n = 4, *Shh^+/–^* n = 4, KA: WT n = 6, *Shh^+/–^* n = 6 mice). #p < 0.05. Unpaired *t*-test (two-tailed) in two groups (WT vs *Shh^+/–^* in naive condition). There was reduction of DCX+ BrdU+ cells between WT vs Shh^+/–^ in naive condition.

The online version of this article includes the following source data and figure supplement(s) for figure 1:

**Source data 1.** Raw data for counts.

**Figure supplement 1.** Sonic hedgehog (Shh)-responding cells in the SGZ are neural stem cells (NSCs).

**Figure supplement 1—source data 1.** Raw data for counts.

amount of *Shh* expression is upregulated by seizures in mossy cells. To address this, we visualized *Shh* mRNA in the DG using RNAscope in situ hybridization technology. Consistent with the results from *Shh^Cre/+^*;Rosa^AI14^ mice, we observed that *Shh* mRNA is enriched in the dentate hilus (**Figure 2E**). Furthermore, *Shh* expression was dramatically increased throughout the dentate hilus from anterior to posterior of dorsal DG at 24 hr after seizure induction. Combining RNAscope with immunofluorescence, we analyzed the number of *Shh* mRNA puncta in the GluR2/3+ mossy cells and found significantly increased *Shh* in mossy cells after seizure induction (**Figure 2F, G**). Taken together, these data support the conclusion that mossy cells are the local source of *Shh* in the DG and that *Shh* mRNA expression is increased upon seizure activity.

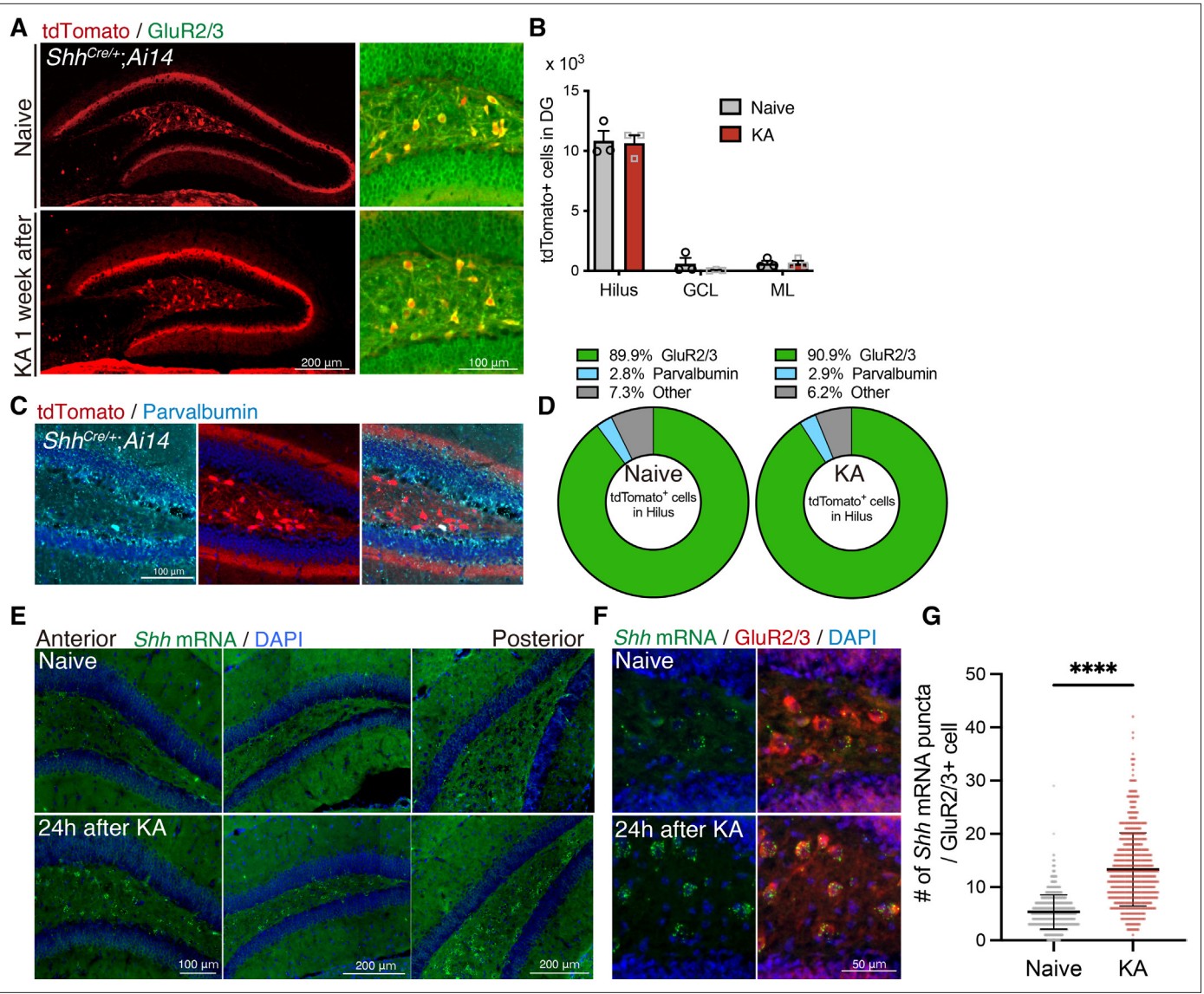

**Figure 2.** Sonic hedgehog (Shh) is expressed in mossy cells and upregulated upon seizure induction. (**A**) Representative immunofluorescence images of mossy cells labeled with GluR2/3 (green) and *Shh^{Cre/+}*;*Rosa^{AI14}* (tdTomato; red) 1 week after seizure induction. (**B**) Quantification of tdTomato+ cells in each area of the dentate gyrus (DG). GCL: granule cell layer, ML: molecular layer (naive: *n* = 3, kainic acid [KA]: *n* = 3 mice). (**C**) Representative immunofluorescence images of interneurons labeled with parvalbumin (cyan) and *Shh^{Cre/+}*;*Rosa^{AI14}* (tdTomato; red) in the hilus. (**D**) Cell-type population of tdTomato+ cells in the hilus of *Shh^{Cre/+}*;*Rosa^{AI14}* mice 1 week after seizure induction (naive: *n* = 3, KA: *n* = 3 mice). (**E**) In situ RNAscope detection of *Shh* mRNA (green) in the DG from anterior to posterior 24 hr after seizure induction. (**F**) Representative RNAscope-immunofluorescence images for *Shh* expression in mossy cells labeled with GluR2/3 (red). (**G**) Quantification of *Shh* mRNA puncta in GluR2/3+ mossy cells. Values represent mean ± standard error of the mean (SEM); ****$p < 0.0001$. Unpaired *t*-test (two-tailed). A total of 608 and 584 GluR2/3+ cells were quantified from three mice in naive and KA treated groups, respectively.

The online version of this article includes the following source data and figure supplement(s) for figure 2:

**Source data 1.** Raw data for counts.

**Figure supplement 1.** *Shh* is expressed in dorsal mossy cells.

## Shh from mossy cells contributes to seizure-induced aberrant neurogenesis

To examine the role of Shh derived from mossy cells in seizure-induced activation of Shh signaling, we generated mossy cell-selective conditional *Shh* knock-out (*Shh*-cKO) mice by crossing a floxed *Shh* allele (*Shh^{fl}*) with calcitonin receptor-like receptor (*Crlr*)-*Cre* mice, in which Cre-mediated recombination

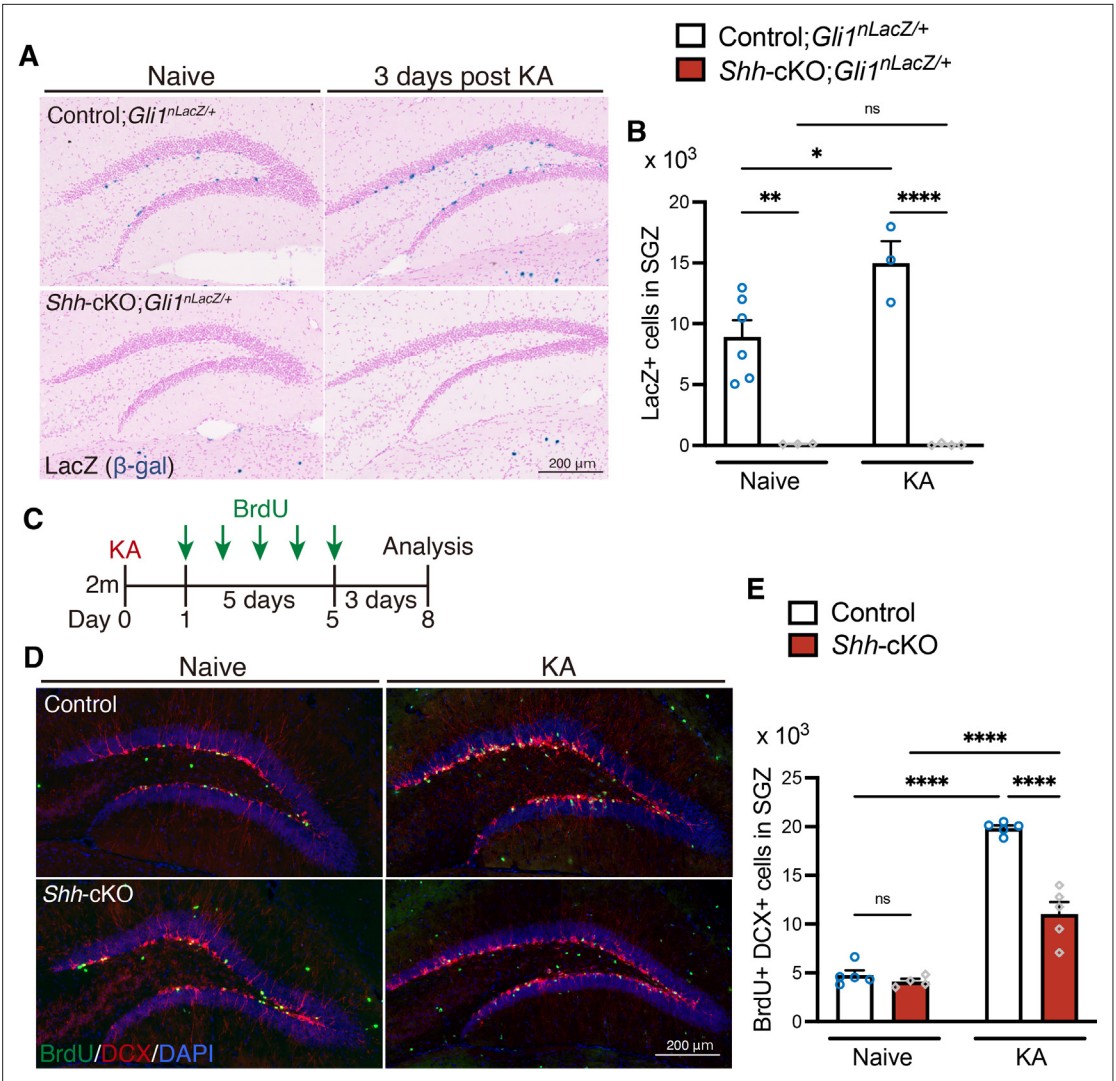

**Figure 3.** Deletion of Sonic hedgehog (Shh) in mossy cells decreases Shh signaling activation and attenuates seizure-induced neurogenesis. (**A**) Representative Gli1-LacZ staining images of the dentate gyrus (DG) of control (*Crlr-Cre;Shh⁺/⁺* and *Shh⁺/⁺*, *Shh ᶠˡ/⁺* and *Shh ᶠˡ/ᶠˡ* without *Cre*), and *Crlr-Cre;Shhᶠˡ/ᶠˡ* (*Shh*-cKO);*Gli1ⁿˡᵃᶜᶻ/⁺* mice 3 days after kainic acid (KA)-induced seizure. (**B**) Quantification of Gli1-nLacZ+ cells in SGZ. Values represent mean ± standard error of the mean (SEM); ns: p > 0.05, *p < 0.05, **p < 0.01, ****p < 0.0001. Two-way analysis of variance (ANOVA) with Tukey's multiple comparison test (naive: control;*Gli1ⁿˡᵃᶜᶻ/⁺* n = 6, *Shh*-cKO; *Gli1ⁿˡᵃᶜᶻ/⁺* n = 3, KA: control;*Gli1ⁿˡᵃᶜᶻ/⁺* n = 3, *Shh*-cKO; *Gli1ⁿˡᵃᶜᶻ/⁺* n = 4 mice). (**C**) Experimental scheme of analyzing neurogenesis by 5-bromo-2'-deoxyuridine (BrdU) pulse labeling. One day after KA-induced seizures, control and *Shh*-cKO mice were injected with BrdU for 5 days and analyzed 3 days after the last BrdU injection. (**D**) Representative immunofluorescence images for newborn neurons labeled with BrdU (green), DCX (red), and 4',6-diamidino-2-phenylindole, dihydrochloride (DAPI; blue) in the SGZ of control and *Shh*-cKO mice after seizure induction (KA). (**E**) Quantification of newborn neurons after seizure induction in the SGZ of control and *Shh*-cKO mice. Values represent mean ± SEM; ns: p > 0.05, ****p < 0.0001. Two-way ANOVA with Tukey's multiple comparison test (naive: control n = 5, *Shh*-cKO n = 4, KA: control n = 5, *Shh*-cKO n = 5 mice).

The online version of this article includes the following source data and figure supplement(s) for figure 3:

**Source data 1.** Raw data for counts.

**Figure supplement 1.** Deletion of *Shh* from mossy cells does not change neural stem cell (NSC) number.

**Figure supplement 1—source data 1.** Raw data for counts.

is induced in mossy cells and a subset of CA3 neurons in the hippocampus (*Jinde et al., 2012*). We first tested whether deletion of *Shh* in mossy cells affects Shh signaling using *Gli1ⁿˡᵃᶜᶻ/⁺* mice. We found that the number of Gli1-nLacZ+ cells was significantly decreased in *Shh*-cKO mice compared with control mice (*Crlr-Cre;Shh⁺/⁺* mice and *Shh⁺/⁺*, *Shhᶠˡ/⁺* and *Shhᶠˡ/ᶠˡ* without the *Cre* transgene were used as controls) (*Figure 3A, B*). Three days after seizure induction, Gli1-nLacZ+ cells were significantly

increased in control mice. However, this increase was not observed in *Shh*-cKO;*Gli1*$^{nLacZ/+}$ mice, in which Gli1-nLacZ+ cells were rarely detected in the SGZ. We found that there were comparable number of Sox2+ GFAP+ radial NSCs in *Shh*-cKO mice compared with control mice (***Figure 3—figure supplement 1***), indicating that reduction of Gli1-nLacZ+ cells is not due to general loss of NSCs. These data suggest that Shh from mossy cells is responsible for Shh signaling in the DG, and that Shh signaling fails to be upregulated by seizures in the absence of Shh from mossy cells.

Since we found impairment in seizure-induced activation of Shh signaling in *Shh*-cKO mice, we next investigated whether Shh from mossy cells is involved in seizure-induced aberrant neurogenesis. The mice were treated with BrdU for 5 days, starting 1 day after seizure induction, and analyzed for the number of BrdU-labeled DCX+ newborn neurons at 3 days after the last BrdU injection (***Figure 3C***). We found that the number of DCX+ BrdU+ newborn neurons was comparable between control and *Shh*-cKO mice in the KA treatment-naive condition, suggesting that deletion of Shh from mossy cells does not significantly affect baseline neurogenesis in this timeframe. After seizure induction, the number of BrdU+ DCX+ newborn neurons was significantly increased in both groups. However, we found that the number of BrdU+ DCX+ newborn neurons was significantly reduced in *Shh*-cKO mice compared to control mice after seizure induction (***Figure 3D, E***), indicating that deletion of *Shh* in mossy cells attenuates seizure-induced neurogenesis. Together, these data indicate that mossy cells function as a critical source of Shh for Shh signaling activity of NSCs, and that Shh derived from mossy cells contributes to seizure-induced neurogenesis.

## Mossy cells control adult neurogenesis though Shh in an activity-dependent manner

A previous study showed that neuronal activity triggers secretion of Shh from hippocampal neurons in culture (***Su et al., 2017***). Since mossy cells undergo hyperactivation during seizures, we hypothesized that once activated by seizures, mossy cells increase secretion of Shh and contribute to seizure-induced aberrant neurogenesis. To test whether neuronal activity in mossy cells increases neurogenesis, we used a chemogenetic approach to activate mossy cells. *Crlr-Cre* mice were crossed with Cre-dependent Designer Receptors Exclusively Activated by Designer Drugs (DREADD) activator mice (*Rosa*$^{DIO-hM3Dq}$) which express excitatory G-protein-coupled receptor hM3Dq with mCherry upon Cre recombination (from here on referred as *Crlr-Cre;hM3Dq*) (***Sciolino et al., 2016***; ***Figure 4A***). In *Crlr-Cre;hM3Dq* mice, mCherry+ recombined cells were observed specifically in the dentate hilus throughout dorsal and ventral hippocampus (***Figure 4B*** and ***Figure 4—figure supplement 1***). As previously reported, we found that over 90% of mCherry+ expressing cells are GluR2/3+ mossy cells (***Jinde et al., 2012***), and that these made up 65% of the GluR2/3+ mossy cells in the dentate hilus (***Figure 4C, D***). Since the high dose of Clozapine-*N*-oxide, frequently used in the DREADD system has a risk of causing behavioral abnormalities and might have contributory effects on neurogenesis (***Gomez et al., 2017***; ***MacLaren et al., 2016***), we instead used Clozapine (CLZ) to induce neuronal activation (***Cho et al., 2020***; ***Gomez et al., 2017***). In addition, we administrated CLZ to both *Rosa*$^{DIO-hM3Dq}$ control and *Crlr-Cre;hM3Dq* mice to normalize the possible side-effects of CLZ administration. 1.5 hr after CLZ injection, we observed that expression of c-fos, a marker of neuronal activation, was significantly increased in the hilus of *Crlr-Cre;hM3Dq* mice (***Figure 4E, F***); we also confirmed that over 70% of GluR2/3+ mCherry+ mossy cells were activated (***Figure 4G***). To investigate the effect of mossy cell neuronal activity on neurogenesis, the mice received BrdU for 5 days concomitant with CLZ treatment (***Figure 4H***) and were analyzed 3 days after the last BrdU injection. We found that the number of DCX+ BrdU+ newborn neurons was significantly increased in *Crlr-Cre;hM3Dq* mice (***Figure 4I, J***), suggesting that increased neuronal activity of mossy cells drove neurogenesis. We therefore next investigated whether Shh signaling activity is activated by mossy cell neuronal activity using *Gli1*$^{nLacZ/+}$ mice. After 6 days of CLZ administration (***Figure 4K***), we found that Gli1-nLacZ+ were significantly increased in the SGZ of *Crlr-Cre;hM3Dq;Gli1*$^{nLacZ/+}$ mice, compared with *Rosa*$^{DIO-hM3Dq}$;*Gli1*$^{nLacZ/+}$ control mice (***Figure 4L, M***), suggesting that Shh signaling activity is upregulated in NSCs by mossy cell neuronal activity. To test the role of Shh in mossy cell neuronal activity-induced neurogenesis, we deleted *Shh* in mossy cells and investigated whether neuronal activity in mossy cells induces neurogenesis using *Crlr-Cre;Shh*$^{fl/fl}$;*hM3Dq* mice. After CLZ administration, we found no induction of neurogenesis by mossy cell neuronal activity in *Crlr-Cre;Shh*$^{fl/fl}$;*hM3Dq* mice (***Figure 4N, O***), indicating that induction of neurogenesis by mossy cell neuronal activity is compromised in the absence of Shh from

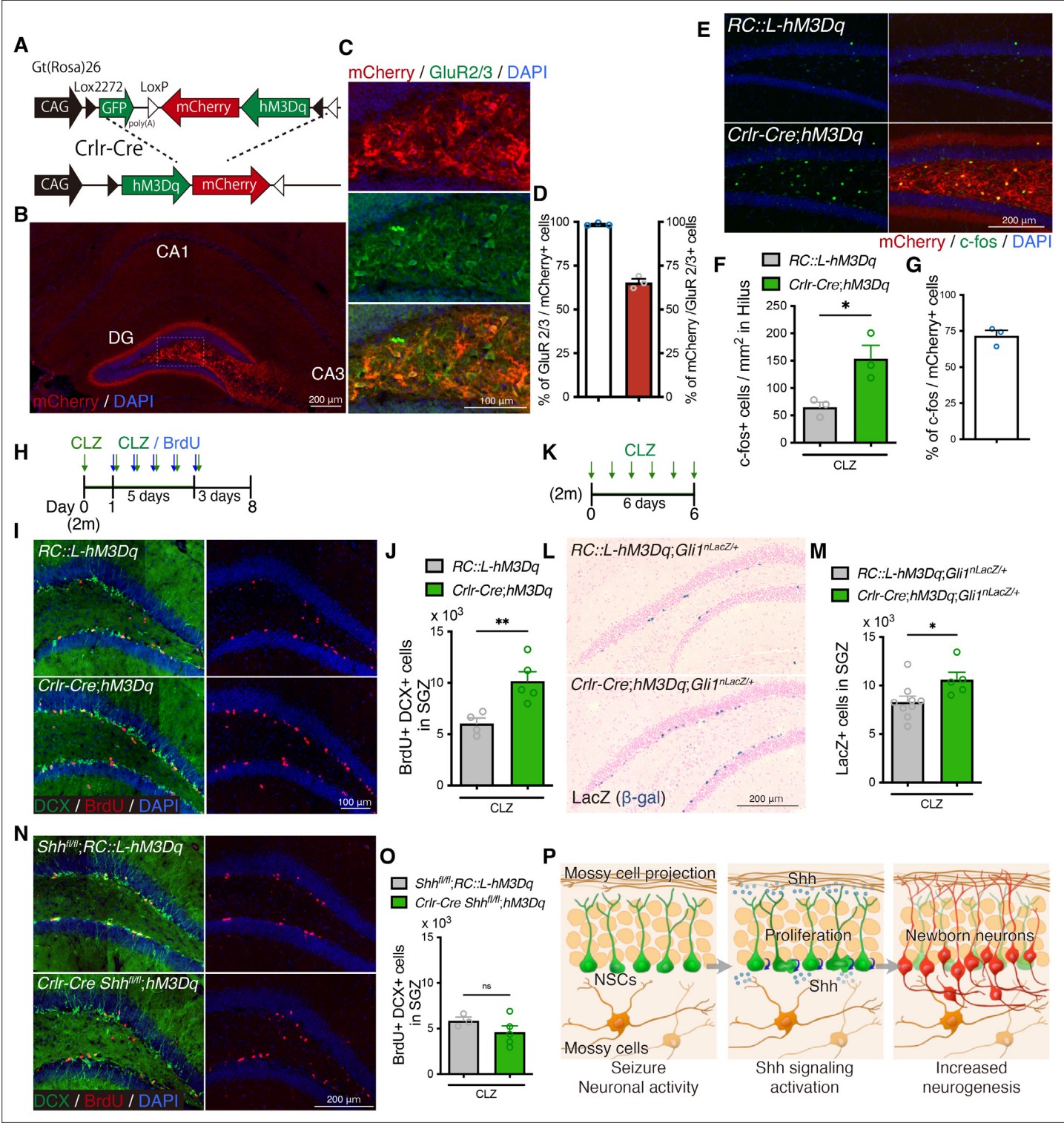

**Figure 4.** Mossy cell neuronal activity induces neurogenesis though Sonic hedgehog (Shh). (**A**) Schematic of conditional expression of Designer Receptors Exclusively Activated by Designer Drugs (DREADD) activator in mossy cells. Representative immunofluorescence images of DREADD activator hM3Dq/mCherry (red) expressing cells in the hippocampus (**B**) and in GluR2/3+ (green) mossy cells of dentate hilus (**C**). (**D**) Quantification of recombination specificity (% of GluR2/3+ mossy cells in mCherry+ recombined cells) and rate (% of mCherry+ recombined cells in GluR2/3+ mossy cells) in GluR2/3+ mossy cells (*n* = 3 mice). Values represent mean ± standard error of the mean (SEM). (**E**) Representative immunofluorescence images of neuronal activation (c-fos, green) in the hilus of *Rosa^DIO-hM3Dq* and *Crlr-Cre;Rosa^DIO-hM3Dq* mice at 1.5 hr after CLZ injection. hM3Dq expressing recombined cells are labeled with mCherry (red). (**F**) Quantification of c-fos+ cells in the hilus of *Rosa^DIO-hM3Dq* and *Crlr-Cre;Rosa^DIO-hM3Dq* mice 1.5 hr after

*Figure 4 continued on next page*

*Figure 4 continued*

CLZ injection. Values represent mean ± SEM; *p < 0.05. Unpaired *t*-test (two-tailed, *Rosa*$^{DIO-hM3Dq}$: *n* = 3, *Crlr-Cre;Rosa*$^{DIO-hM3Dq}$: *n* = 3 mice). (**G**) Induction rate of neuronal activation in the hM3Dq expressing recombined cells (% of c-fos+ cells in mCherry+ cells in the hilus, *n* = 3 mice). Values represent mean ± SEM. (**H**) Experimental scheme of analyzing neurogenesis by 5-bromo-2'-deoxyuridine (BrdU) pulse labeling during DREADD activation. CLZ was administered to 2-month-old (2 m) mice for 6 days. Starting on the second day of CLZ administration, mice concomitantly received BrdU for 5 days and were analyzed 3 days after the last BrdU injection. (**I**) Representative immunofluorescence images for DCX+ (green) BrdU+ (red) newborn neurons produced after DREADD-induced neuronal activation of mossy cells in the SGZ of *Rosa*$^{DIO-hM3Dq}$ and *Crlr-Cre;Rosa*$^{DIO-hM3Dq}$ mice. (**J**) Quantification of newborn neurons in the SGZ. Values represent mean ± SEM; **p < 0.01. Unpaired *t*-test (two-tailed, *Rosa*$^{DIO-hM3Dq}$: *n* = 4, *Crlr-Cre;Rosa*$^{DIO-hM3Dq}$: *n* = 5 mice). (**K**) Experimental scheme of analyzing Shh signaling activation after induction of mossy cell neuronal activity. (**L**) Representative Gli1-nLacZ staining images of the dentate gyrus (DG) of *Rosa*$^{DIO-hM3Dq}$ and *Crlr-Cre;Rosa*$^{DIO-hM3Dq}$ mice after CLZ administration for 6 days. (**M**) Quantification of Gli1-nLacZ+ cells in SGZ. Values represent mean ± SEM; *p < 0.05. Unpaired *t*-test (two-tailed, *Rosa*$^{DIO-hM3Dq}$;*Gli1*$^{nLacZ/+}$: *n* = 9, *Crlr-Cre;Rosa*$^{DIO-hM3Dq}$;*Gli1*$^{nLacZ/+}$: *n* = 5 mice). (**N**) Representative immunofluorescence images for newborn neurons produced after DREADD-induced neuronal activation of mossy cells in the SGZ of *Shh*$^{fl/fl}$;*Rosa*$^{DIO-hM3Dq}$ and *Crlr-Cre;Shh*$^{fl/fl}$;*Rosa*$^{DIO-hM3Dq}$ mice, which are labeled with DCX (green), BrdU (red), and 4',6-diamidino-2-phenylindole, dihydrochloride (DAPI; blue). (**O**) Quantification of newborn neurons in the SGZ of *Shh*$^{fl/fl}$ *Rosa*$^{DIO-hM3Dq}$ and *Crlr-Cre;Shh*$^{fl/fl}$;*Rosa*$^{DIO-hM3Dq}$ mice after DREADD-induced neuronal activation of mossy cells. Values represent mean ± SEM; ns: *P*>0.05. Unpaired *t*-test (two-tailed, *Shh*$^{fl/fl}$;*Rosa*$^{DIO-hM3Dq}$: *n* = 3, *Crlr-Cre Shh*$^{fl/fl}$;*Rosa*$^{DIO-hM3Dq}$: *n* = 5 mice). (**P**) Our proposed model of mossy cell-mediated neurogenesis after seizures. *Shh* expression in mossy cells are upregulated by seizure activity. Neuronal activity of mossy cells activates Shh signaling in neural stem cells (NSCs) and contributes to the seizure-induced neurogenesis.

The online version of this article includes the following source data and figure supplement(s) for figure 4:

**Source data 1.** Raw data for counts.

**Figure supplement 1.** *Crlr-Cre*-mediated recombination in the dentate gyrus (DG).

**Figure supplement 2.** Neuronal activity of mossy cells induces neurogenesis in the contralateral dentate gyrus (DG).

**Figure supplement 2—source data 1.** Raw data for counts.

**Figure supplement 3.** Deletion of *Shh* attenuates seizure-induced neurogenesis in the contralateral dentate gyrus (DG).

**Figure supplement 3—source data 1.** Raw data for counts.

mossy cells. Together, these data indicate that Shh from mossy cells contributes to neuronal activity-induced neurogenesis by mossy cells.

Since mossy cells project their axons to the contralateral DG (*Botterill et al., 2021*; *Houser et al., 2021*), we then examined whether mossy cell neuronal activity influences neurogenesis in the contra-lateral DG. To address this, we stereotaxically injected Cre-dependent adeno-associated virus (AAV) expressing the excitatory G-protein-coupled receptor hM3Dq-mCherry under the human synapsin promoter (AAV::hSyn-DIO-hM3Dq-mCherry, from here on referred as AAV::hM3Dq-mCherry) into the DG of *Crlr-Cre* mice (*Figure 4—figure supplement 2A*). The mice were treated with CLZ to activate hM3Dq and analyzed 1.5 hr after CLZ injection. We found the specific infection of dorsal mossy cells in the AAV-injected side of the DG and contralateral projection of infected mossy cells in the inner and middle molecular layer in the contralateral DG (*Figure 4—figure supplement 2B–D*). We also confirmed that CLZ injection increases the expression of c-fos in the AAV::hM3Dq-mCherry infected mossy cells of the DG on the injected side (*Figure 4—figure supplement 2E–G*). To test whether mossy cell neuronal activity induces neurogenesis in contralateral DG, we injected AAV virus at 4–5 weeks old, and after 4 weeks recovery, mice received BrdU for 5 days concomitant with CLZ treatment and were analyzed 3 days after the last BrdU injection (*Figure 4—figure supplement 2H*). We found that the number of newborn neurons was significantly increased on the contralateral DG of AAV::hM3Dq-mCherry-injected mice (*Figure 4—figure supplement 2I, J*), indicating that mossy cell neuronal activity contributes to increasing neurogenesis in the contralateral DG.

Previous studies showed that Shh can be secreted from axons and anterogradely transported to distal regions along the axon (*Beug et al., 2011*; *Peng et al., 2018*; *Su et al., 2017*). These reports, together with our findings, raise the possibility that increased *Shh* expression after seizures is trans-ported to the contralateral DG, thereby allowing mossy cells to regulate contralateral neurogenesis with neuronal activity. To address this possibility, we deleted *Shh* in mossy cells in a unilateral hemi-sphere of DG by injecting AAV::Cre-P2A-tdTomato (AAV::Cre-tdTomato) into the DG of *Shh*$^{fl/fl}$ mice at 4–5 weeks old and investigated seizure-induced neurogenesis in the contralateral DG in 2-month-old mice (*Figure 4—figure supplement 3A*). We found that infection by AAV::Cre-tdTomato was restricted to the injected DG. Dorsal mossy cells send projections to the inner and middle molecular layers of DG bilaterally into both hemispheres along the septo-temporal axis (*Botterill et al., 2021*; *Houser et al.,*

*2021*). We found tdTomato+ mossy cell projections in the contralateral DG from the DG ipsilateral to the injection site (*Figure 4—figure supplement 3B*), indicating that AAV::Cre-tdTomato infected mossy cells. *Shh* expression is observed predominantly in dorsal mossy cells (*Figure 2—figure supplement 1*). We observed AAV::Cre-tdTomato infection in 31% of total GluR2/3+ mossy cells and 45% of GluR2/3+ dorsal mossy cells in the DG ipsilateral to the injection site (*Figure 4—figure supplement 3C, D*). We then tested whether *Shh* deletion affects contralateral neurogenesis (*Figure 4—figure supplement 3E*) and found that seizure-induced neurogenesis was attenuated in the contralateral DG of mice injected with AAV::Cre-tdTomato, in which DCX+ BrdU+ newborn neurons induced by seizure activity were significantly reduced compared with mice injected with AAV::mCherry control virus (*Figure 4—figure supplement 3F, G*). This suggests that mossy cells control contralateral neurogenesis by regulating Shh expression after seizures. Together with our findings that *Shh* expression was increased by seizure (*Figure 2F, G*), these data demonstrate that increased Shh following seizures may be transported and contribute to neurogenesis in the contralateral DG. Taken together, these data support our hypothesis that mossy cells release Shh with increased neuronal activity and induce neurogenesis driven by seizures (*Figure 4P*).

## Deletion of *Shh* in mossy cells leads to reduction of the NSC pool after seizure-induced neurogenesis

Whether there is a functional benefit of increased *Shh* expression in mossy cells following seizure activity remains unclear, considering that seizure-induced aberrant neurogenesis negatively affects hippocampal function. Recently, it was shown that a population of quiescent NSCs in the adult DG, once activated to proliferate and produce neurons, can return to quiescence rather than being consumed by terminal differentiation (*Botterill et al., 2021*; *Harris et al., 2021*). This mechanism is suggested to contribute to the long-term persistence of NSCs and maintenance of neurogenesis throughout life by decelerating NSC pool exhaustion. Since Shh is important for proliferation and self-renewal of NSCs (*Ahn and Joyner, 2005*; *Choe and Pleasure, 2013*; *Lai et al., 2003*), we investigated whether Shh from mossy cells is involved in maintenance of NSCs after proliferation. We labeled proliferating NSCs with BrdU in the drinking water for 5 days after seizure induction (*Figure 5A*). Four weeks after BrdU administration, we analyzed the number of NSCs retaining BrdU, as previously addressed (*Harris et al., 2021*). Proliferated NSCs labeled by BrdU are eliminated from the NSC pool once they differentiate into neurons, whereas NSCs that have returned to the stem cell state after proliferation will be detected as BrdU-retaining Sox2+ GFAP+ radial NSCs even 4 weeks after proliferation (*Figure 5B*). We found that seizure induction increases the number of BrdU+ Sox2+ proliferating NSCs at day 0 (last day of BrdU labeling) in both control and *Shh*-cKO mice compared with the no seizure condition, although it was significantly attenuated in *Shh*-cKO mice compared with control mice (*Figure 5C, D*). Four weeks after BrdU labeling, we found that the number of BrdU-retaining Sox2+ GFAP+ radial NSCs was increased in seizure-induced mice compared with no seizure-induced mice, as expected from the increased proliferation after seizure (*Figure 5C, E*). However, the number of BrdU-retaining Sox2+ GFAP+ radial NSCs in *Shh*-cKO mice was significantly reduced compared with seizure-induced control mice (*Figure 5C, E*). To assess the persistence of NSCs' identity after seizure-induced proliferation, we calculated the fraction of NSCs that persist as NSCs 4 weeks after proliferation by dividing the number of BrdU-retaining Sox2+ GFAP+ radial NSCs at 4 weeks with the number of BrdU+ Sox2+ proliferating NSCs at day 0 (*Figure 5F*). We found that the fraction of NSCs that retain stem cell identity after proliferation was comparable in control mice with and without seizure induction. This indicates that even though seizures induce NSCs to proliferate, they return to the stem cell state in the same ratio (*Figure 5F*). In contrast, we found that the fraction of NSCs that returned to the stem cell state in *Shh*-cKO mice was significantly reduced after seizure-induced proliferation (*Figure 5F*), suggesting that NSCs are less likely to return to their stem cell state after seizure-induced proliferation in the absence of Shh from mossy cells.

Seizures increase NSC proliferation and subsequent neurogenesis in the short term. However, it has been suggested that increased neurogenesis induced by neuronal hyperactivity accelerates consumption of NSCs and leads to depletion of NSC pool over time (*Fu et al., 2019*; *Sierra et al., 2015*). Since NSCs in *Shh*-cKO mice had fewer NSCs that returned to their stem cell state after seizure-induced proliferation, we next investigated the consequences on the NSC pool after seizure-induced neurogenesis. We analyzed the number of Sox2+ GFAP+ radial NSCs 4 weeks after BrdU labeling (which is

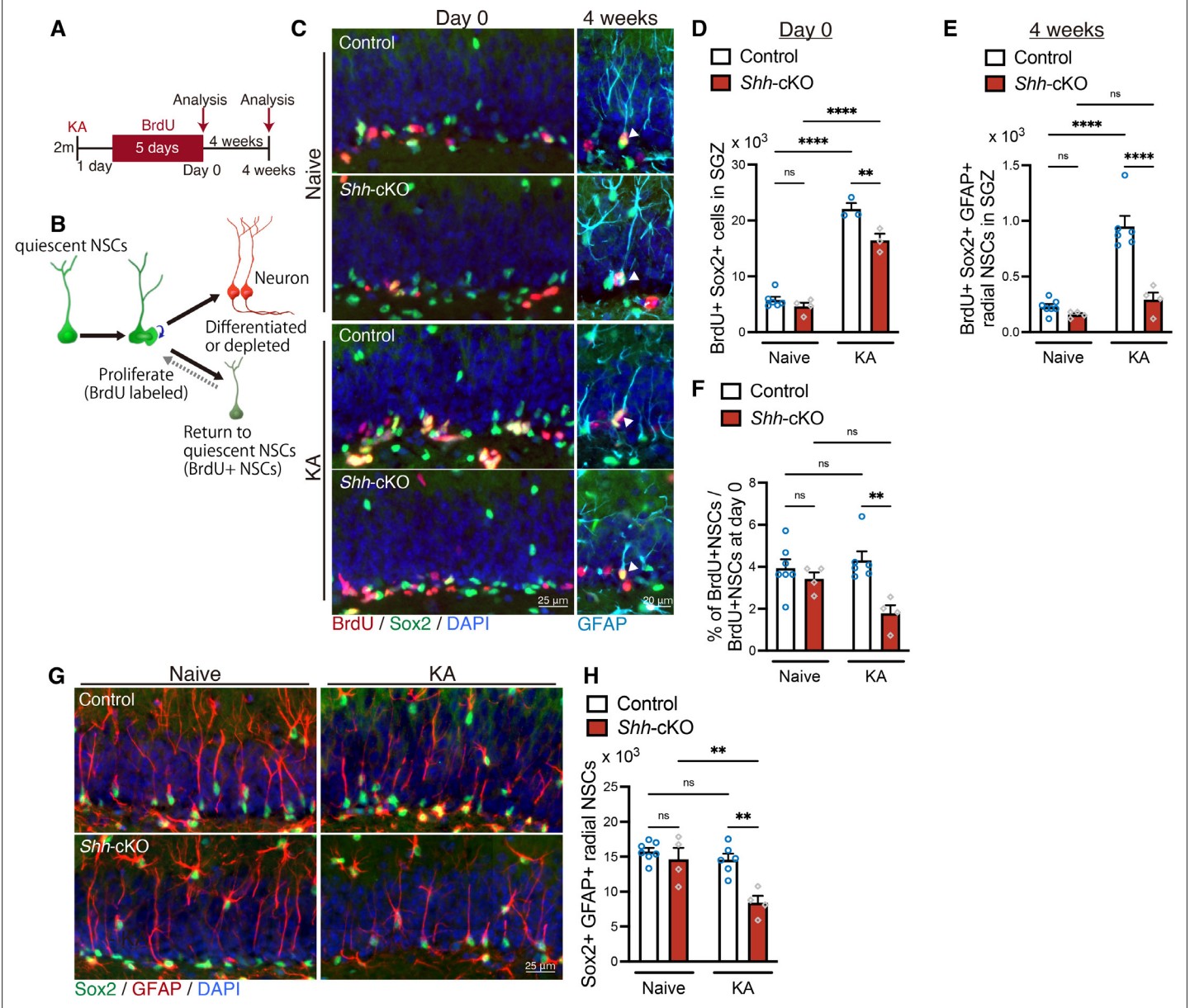

**Figure 5.** Sonic hedgehog (Shh) from mossy cells maintains neural stem cell (NSC) pool after seizure-induced neurogenesis. (**A**) Experimental scheme of long-term tracing of NSCs that proliferate after seizure induction. Seizures were induced in 2-month-old (2 m) control and *Shh*-cKO mice by kainic acid (KA) injections. One day after seizure induction, mice received 5-bromo-2′-deoxyuridine (BrdU) via drinking water for 5 days and were analyzed on the last day of BrdU administration (day 0) and at 4 weeks after BrdU administration. (**B**) Schematic of NSC fate after proliferation. (**C**) Representative immunofluorescence images of proliferating Sox2+ (green) NSCs labeled with BrdU (red) on day 0 and radial GFAP+ (cyan) Sox2+ (green) NSCs retaining BrdU (red) at 4 weeks after BrdU administration, which are indicated by white arrowheads. (**D**) Quantification of Sox2+ BrdU+ proliferating NSCs after seizure induction on day 0. Values represent mean ± standard error of the mean (SEM); ns: $p > 0.05$, **$p < 0.01$, ****$p < 0.0001$. Two-way analysis of variance (ANOVA) with Tukey's multiple comparison test (naive: control $n = 6$, *Shh*-cKO $n = 4$, KA: control $n = 3$, *Shh*-cKO $n = 3$ mice). (**E**) Quantification of Sox2+ GFAP+ radial NSCs retaining BrdU at 4 weeks after BrdU administration. Values represent mean ± SEM; ns: $p > 0.05$, ****$p < 0.0001$. Two-way ANOVA with Tukey's multiple comparison test (naive: control $n = 7$, *Shh*-cKO $n = 4$, KA: control $n = 6$, *Shh*-cKO $n = 4$ mice). (**F**) Quantification of NSC persistence 4 weeks after BrdU administration in naive and KA-induced mice. Values represent mean ± SEM; ns: $p > 0.05$, **$p < 0.01$. Two-way ANOVA with Tukey's multiple comparison test (naive: control $n = 7$, *Shh*-cKO $n = 4$, KA: control $n = 6$, *Shh*-cKO $n = 4$ mice). (**G**) Representative immunofluorescence images of Sox2+ (green) GFAP+ (red) radial NSCs at 4 weeks after BrdU administration (5 weeks after KA-induced seizure). (**H**) Quantification of Sox2+ GFAP+ radial NSCs at 5 weeks after seizure induction in control and *Shh*-cKO mice. Values represent mean ± SEM; ns: $p > 0.05$, **$p < 0.01$. Two-way ANOVA with Tukey's multiple comparison test (naive: control $n = 7$, *Shh*-cKO $n = 4$, KA: control $n = 6$, *Shh*-cKO $n = 4$ mice).

The online version of this article includes the following source data for figure 5:

*Figure 5 continued on next page*

*Figure 5 continued*
**Source data 1.** Raw data for counts.

5 weeks after seizure induction). We found that the number of Sox2+ GFAP+ radial NSCs in the SGZ of control mice is comparable with and without seizures, suggesting that the NSC pool size is maintained at least 5 weeks after seizure induction in control mice (*Figure 5G, H*). However, in *Shh*-cKO mice, the number of Sox2+ GFAP+ radial NSCs was significantly reduced 5 weeks after seizure induction (*Figure 5G, H*), suggesting that deletion of *Shh* in mossy cells leads to premature depletion of the NSC pool after seizure-induced neurogenesis. Taken together, these data imply that Shh from mossy cells is required for maintaining the NSC pool after seizure-induced neurogenesis, and that Shh from mossy cells contribute to the NSCs' return to their stem cell state after seizure-induced proliferation, thus preventing NSC pool depletion.

### Loss of *Shh* from mossy cells accelerates age-related decline of the NSC pool

NSCs in the adult DG self-renew and return to a quiescent state after proliferation more often in aged mice than in young mice, which is thought to be the mechanism for maintaining the NSC pool during the aging process (*Harris et al., 2021*). We therefore next investigated whether deletion of *Shh* in mossy cells influences the aging-related NSC pool decline using 9- to 11-month-old aged mice. We found that the number of Sox2+ GFAP+ radial NSCs was significantly reduced in *Shh*-cKO mice compared with control mice (*Figure 6A, B*). We next asked whether the aging-related increase in NSC self-renewal is compromised in *Shh*-cKO mice. We administrated BrdU to mice via drinking water for 5 days to label proliferating cells and investigated the number of NSCs retaining BrdU in the SGZ at 10 days and 4 weeks after BrdU administration (*Figure 6C*). Consistent with previous reports (*Harris et al., 2021*), we found that a greater fraction of NSCs retained BrdU until 4 weeks after BrdU labeling in aged control mice compared with 2-month-old mice (*Figures 5C, F, 6D, E*), indicating that NSCs in aged mice tend to retain their stem cell state after proliferation compared to younger mice. However, in *Shh*-cKO mice, the fraction of NSCs retaining BrdU after BrdU labeling was significantly reduced compared with control mice (*Figure 6D, E*), suggesting that NSCs in aged *Shh*-cKO mice are less able to retain their stem cell state after proliferation. Together, our data suggest that Shh from mossy cells contributes to persistence of the NSC state after proliferation throughout life, which sustains the NSC pool during aging.

## Discussion

Here, we showed that hilar mossy cells are the source of Shh in the DG, and that *Shh* expression and signaling activity in the DG are increased by seizure activity and contribute to seizure-induced neurogenesis. Deletion of *Shh* from mossy cells attenuated the typical induction of neurogenesis by seizures. However, it also led to the reduction of the NSC pool after seizure-induced neurogenesis. Based on our results, we propose a hypothetical model for NSC pool regulation by mossy cells, in which hilar mossy cells provide Shh, which is important for NSC proliferation and self-renewal to maintain the NSC pool, thereby preventing excessive consumption of NSCs after the increased NSC proliferation that occurs during seizure activity.

We found that seizure-induced neurogenesis was attenuated in *Shh*-cKO mice, in which Shh signaling activity was diminished and failed to be activated upon seizure. However, neurogenesis in *Shh*-cKO mice was still significantly increased by seizure induction. This suggests that there must be additional signaling molecules and factors inducing neurogenesis following seizures in addition to Shh. The canonical Wnt signaling pathway is also upregulated after seizures and has also been suggested to be important in seizure-induced neurogenesis (*Madsen et al., 2003*; *Mardones and Gupta, 2022*; *Qu et al., 2017*). KA injection activates proliferation of both NSCs and intermediate progenitors (*Lugert et al., 2010*; *Sierra et al., 2015*). Our previous findings demonstrated that activation of Wnt signaling contributes to both NSCs and intermediate progenitor expansion whereas Shh signaling activity increases NSC proliferation (*Choe and Pleasure, 2013*). Thus, these signaling pathways may function together to induce and regulate neurogenesis following seizures, and Shh from mossy cells may contribute to proliferation of NSCs during seizure activity.

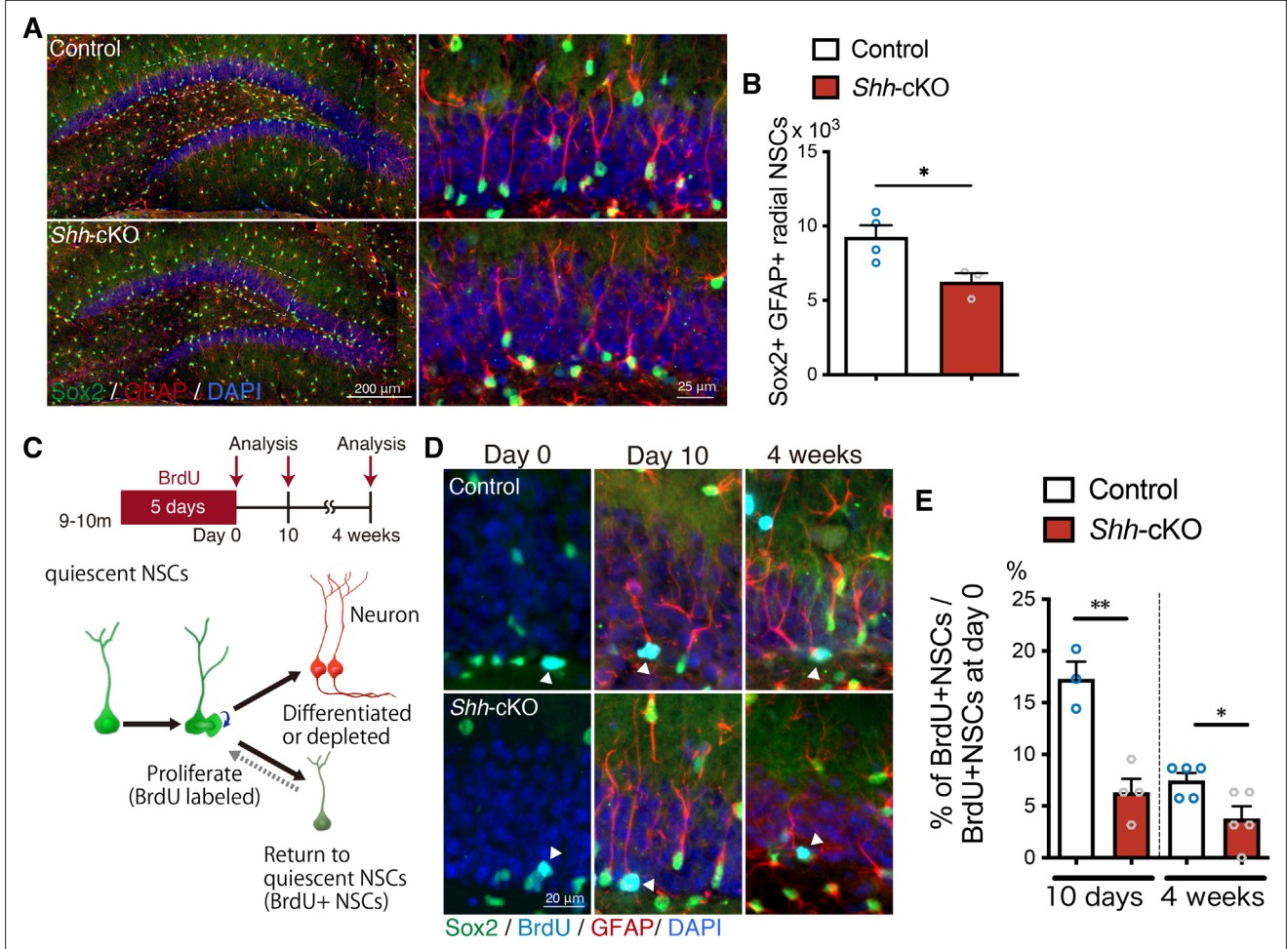

**Figure 6.** Deletion of *Shh* in mossy cells increases age-related NSC decline. (**A**) Representative immunofluorescence images of Sox2+ (green) GFAP+ (red) radial NSCs in 9- to 11-month-old control and *Shh*-cKO mice. High magnification of inset images are shown in the right panels. (**B**) Quantification of Sox2+ GFAP+ radial NSCs. Values represent mean ± standard error of the mean (SEM); *p < 0.05. Unpaired *t*-test (two-tailed, control: n = 4, *Shh*-cKO: n = 3 mice). (**C**) Experimental scheme of long-term tracing of proliferating NSCs in aged mice. Nine- to eleven-month-old control and *Shh*-cKO mice were administrated with 5-bromo-2'-deoxyuridine (BrdU) via drinking water for 5 days and were analyzed on the last day of BrdU administration (day 0), and at 10 days and 4 weeks after BrdU administration. Schematic of NSC fate after proliferation. (**D**) Representative immunofluorescence images of proliferating Sox2+ (green) NSCs labeled with BrdU (cyan) at day 0 and radial Sox2+ (green) GFAP+ (red) NSCs retaining BrdU (cyan) at 10 days and 4 weeks after BrdU administration. (**E**) Quantification of the fraction of NSCs retaining BrdU until 10 days and 4 weeks after BrdU administration in aged control and *Shh*-cKO mice. Values represent mean ± SEM; *p < 0.05, **p < 0.01. Unpaired *t*-test (two-tailed, 10 days: control: n = 3, *Shh*-cKO: n = 4, 4 weeks: control: n = 5, *Shh*-cKO: n = 5 mice mice).

The online version of this article includes the following source data and figure supplement(s) for figure 6:

**Source data 1.** Raw data for counts.

**Figure supplement 1.** Expressing Designer Receptors Exclusively Activated by Designer Drugs (DREADD) activator does not affect neurogenesis and number of NSCs of *Crlr-Cre Shh* mice.

**Figure supplement 1—source data 1.** Raw data for counts.

Previous research has demonstrated that mossy cell neuronal activity increases NSC proliferation through direct excitatory glutamatergic signaling onto NSCs (*Yeh et al., 2018*). Interestingly, Shh has been shown to inhibit the activity of glutamate transporters and increase extracellular glutamate (*Feng et al., 2016*). This raises the possibility that in addition to increasing Shh signaling activity in NSCs, Shh from mossy cells may also contribute to increasing glutamatergic signaling onto NSCs. Our results show that neurogenesis induced by mossy cell neuronal activity was compromised in *Shh*-cKO mice. Considering this together with our findings, this may be the result of concomitant reduction of Shh signaling activity and glutamatergic signaling onto NSCs in the absence of Shh from mossy cells.

Whether Shh from mossy cells influences glutamatergic signaling to NSCs needs further investigation. However, the combination of increased Shh signaling and glutamatergic signaling activity by Shh from mossy cells may be an additional mechanism for mossy cell-mediated neurogenesis with seizures.

Previous studies indicated that NSCs in the adult DG proliferate multiple times once they are activated, and that NSCs display self-renewal through both symmetric and asymmetric cell division (*Bonaguidi et al., 2011*; *Bottes et al., 2021*; *Harris et al., 2021*). How the number and type of cell divisions of adult NSCs is regulated remains elusive. However, there are several findings suggesting that Shh signaling activity is involved in symmetric self-renewal and in regulating the number of cell divisions. In vivo live imaging of NSCs in adult DG using two different *Cre* lines for labeling NSCs demonstrated that NSCs undergo symmetric self-renewal only at the first cell division (*Bottes et al., 2021*). Interestingly, in their study, *Gli1-CreER*-tagged NSCs show long-term self-renewal capacity, and symmetric self-renewal was observed more abundantly in *Gli1-CreER*-tagged NSCs than in *Ascl1-CreER*-tagged NSCs, suggesting that Shh-responding NSCs have robust symmetric self-renewal capacity (*Bottes et al., 2021*). In chick spinal cord development, high Shh signaling activity has been shown to increase symmetric self-renewal of neuroepithelial cells (*Saade et al., 2017*). Although whether similar mechanism is adapted to regulate NSC proliferation in adult DG needs to be elucidated, this raises the possibility that increased Shh from mossy cells following seizures promotes the self-renewal of NSCs though symmetric cell division and contribute to maintaining the NSC pool. Furthermore, a study pulse-labeling NSCs by H2B-GFP demonstrated that NSCs in 6-month-old mice undergo a greater number of cell divisions compared with 1.5-month-old juvenile mice (*Harris et al., 2021*). Single-cell RNA sequencing results showed that direct Shh signaling target genes, *Gli1* and *Ptch1* expression are enriched in NSCs of aged mice compared to young mice (*Harris et al., 2021*). Thus, there is a possibility that Shh signaling activity is elevated in aged NSCs, which may contribute to self-renewal and increase of cell division with age. Given this, the reduced self-renewal and number of cell divisions may be a plausible explanation for the decreased ratio of NSCs that retain stem cell state in *Shh*-cKO mice after seizure induction and aging. Our data showed that the number of NSCs was significantly reduced in *Shh*-cKO aged mice, whereas it was comparable with control mice at 2 months of age. Accordingly, we found that NSCs were less likely to return to their stem cell state in aged, but not 2-month-old *Shh*-cKO mice. These observations suggest that Shh from mossy cells becomes especially critical for maintaining the NSC pool with age. NSC pool reduction in *Shh*-cKO mice may result from accumulated impairment of NSC self-renewal over time. Together, our results support the conclusion that Shh signaling activity is important for maintaining the NSC pool during both seizure and aging, and that mossy cell-derived Shh promotes long-term stem cell potency of NSCs.

Previous studies showed that mossy cells receive inputs intrinsic and extrinsic to the hippocampus and are involved in hippocampus-dependent behaviors (*Azevedo et al., 2019*; *Scharfman, 2016*; *Sun et al., 2017*; *Wang et al., 2021*). There is great interest in whether mossy cell-mediated neurogenesis is involved in other forms of activity that can induce neurogenesis, such as exercise and environmental enrichment (*Kempermann, 2008*; *van Praag et al., 1999b*). Whether neurogenesis induced by these types of activity leads to depletion of the NSC pool requires further investigation. We believe that mossy cell-derived Shh-mediated NSC regulation provides new insight into the mechanisms regulating the maintenance of the NSC pool while also increasing neurogenesis. In addition, there is evidence that Alzheimer's disease model mice with seizures have premature deletion of the NSC pool after increased neurogenesis (*Fu et al., 2019*). Our findings provide a potential role for Shh to prevent the depletion of NSC after seizures, in which mossy cells function as a major source of Shh in the DG. Thus, modulation of mossy cell *Shh* expression in neurodegenerative diseases may be a therapeutic approach for maintaining the NSC pool and preserving neurogenesis during neurodegenerative diseases. Our findings contribute important new insights to further understand how seizures and aging lead to dysregulation in NSC behaviors.

## Materials and methods
### Mice
All mice used in this study were maintained on a 12-hr light/dark cycle with free access to food and water. Mouse colonies were maintained at University of California San Francisco (UCSF) in accordance with National Institutes of Health and UCSF guidelines. The following mouse lines were obtained from

Jackson Laboratory (Bar Harbor, Maine): *Gli1*<sup>*CreERT2/+*</sup> (stock #007913, RRID:MGI:3053957), *Rosa*<sup>*Al14*</sup> (Stock #007908, RRID:MGI:J:155793), *Gli1*<sup>*nLacZ/+*</sup> (stock #008211, RRID:MGI:J:79392), *Shh*<sup>*EGFP-Cre/+*</sup> (stock #005622, RRID:MGI:3053959), *Shh*<sup>*flox/flox*</sup> (stock #004293, RRID:MGI:1934268), and *Crlr-Cre* (stock #023014, RRID:MGI: 5523525). *Rosa-CAG-LoxP2272-GFP-LoxP-mCherry-hM3Dq-LoxP2272-LoxP* (*Rosa*<sup>*DIO-hM3Dq*</sup>) mice were generously provided by Dr. Zachary Knight (University of California, San Francisco, also available from Jackson laboratory, stock # 026943, RRID:MGI: 5771785). All mice were randomly assigned to experiments. Both male and female mice were analyzed with no distinction, except for seizure induction experiments, where only male mice were used. There was no difference in the number of newborn neurons and radial NSCs between *Crlr-Cre;Shh*<sup>*fl/fl*</sup> and *Crlr-Cre;Shh*<sup>*fl/fl*</sup>;*Rosa*<sup>*DIO-hM3Dq*</sup> at 2 and 9–11 months old without clozapine administration (*Figure 6—figure supplement 1*). Thus, in the study of long-term tracing of NSCs in aged mice, both *Crlr-Cre;Shh*<sup>*flox/flox*</sup> and *Crlr-Cre;Shh*<sup>*flox/flox*</sup>;*Rosa*<sup>*DIO-hM3Dq*</sup> mice were used as *Shh*-cKO mice.

## Stereotaxic injection of AAV

One-month-old mice were anesthetized with Isoflurane (Piramal) in the sealed box and placed on stereotaxic instrument (Kopf). Anesthesia was maintained during surgery using a veterinary vaporizer (Surgivet). Unilateral stereotaxic injections were performed into the right dorsal dentate hilus (antero-posterior: −2.0 mm from Bregma, mediolateral: 1.9 mm, dorsoventral: 2.25 mm). All injections were performed using a Nanojector III (Drummond Scientific) with a glass capillary, pre-pulled by vertical micropipette puller (Sutter Instrument company, P-30). 500 nl of virus was injected at a rate of 60 nl/min. After the virus was delivered, the glass capillary was left in place for another 5 min to allow for diffusion of the virus away from the needle tract and was then slowly withdrawn. The mice recovered in a warm cage after suturing of the incision and then returned to their home cages. After 3–4 weeks of recovery, mice were subjected to the experiments at 2 months old. For *Shh* deletion experiments, either AAV-hSyn-mCherry (Addgene, 114472-AAV5) or AAV-hSyn-Cre-P2A-dTomato (Addgene, 107738-AAV5) were injected at a titer of $3.0 \times 10^{11}$ GC/ml. For chemogenetic activation of mossy cell experiments, either AAV-hSyn-DIO-mCherry (Addgene, 50459-AAV8) or AAV-hSyn-DIO-hM3D(Gq)-mCherry (Addgene, 44361-AAV8) were injected at a titer of $4.0 \times 10^{12}$ GC/ml.

## Seizure induction by KA administration

Seizure induction was performed by consecutive intraperitoneal injection of low-dose KA (Enzo Life Sciences, dissolved in saline (0.9% NaCl)) in 2-month-old male mice. Mice received KA initially at a dose of 5 mg/kg. Then 20 min after initial injection, the mice consecutively received 2.5 mg/kg KA injection every 20 min until an initial Racine stage 4/5 seizure was observed, which are defined as rearing with forelimb clonus (stage 4); rearing and falling with forelimb clonus (generalized motor convulsions, stage 5) (*Racine, 1972*). Mice that did not display stage 4/5 seizure after receiving total 30 mg/kg of KA were excluded from the experiments. Seizure induction and behavior measurement were performed blindly to the genotype. Mice receiving no KA treatment were used as naive mice controls.

## Tamoxifen, BrdU, and clozapine administration

Tamoxifen (Sigma) was dissolved in corn oil with 10% of ethanol at 50 mg/ml. *Gli1*<sup>*CreER/+*</sup>;*Rosa*<sup>*Al14*</sup> mice were orally administered 5 mg of tamoxifen once a day for 3 days at 2 months old and analyzed 1 day after last tamoxifen injection. For analyzing neurogenesis by BrdU pulse labeling, mice were intraperitoneally injected with BrdU (Sigma) dissolved in saline at a dose of 50 mg/kg once a day for 5 days at 2 months old and analyzed 3 days after last BrdU injection. For long-term tracing of BrdU-labeled NSCs in the experiments for NSC persistence analysis, mice were provided 1 mg/ml of BrdU in the drinking water for 5 days at 2 or 9- to 10 months old and analyzed on the last day of BrdU administration (day 0), or 10 days or 4 weeks after BrdU administration. For induction of neuronal activity by DREADD-based chemogenetic tools, mice were administrated with clozapine (CLZ, Sigma) by intraperitoneal injection at a dose of 0.1 mg/kg once a day and by drinking water (0.1 mg/100 ml) for 6 days. CLZ drinking water was replaced every 2 days. In the experiments testing induction of neuronal activity by c-fos expression, mice were perfused at 1.5 hr after intraperitoneal injection of 0.1 mg/kg of CLZ. For the analysis of Shh-responding cells using *Gli1*<sup>*nLacZ/+*</sup> mice in DREADD experiments, mice were given CLZ for 6 days and perfused on the last day of CLZ administration. To investigate neurogenesis in

DREADD experiments, mice were given CLZ for 6 days, and starting on the second day of CLZ administration, mice were also given 50 mg/kg of BrdU by intraperitoneal injection once a day for 5 days and perfused 3 days after the last BrdU injection.

## Tissue preparation

The mice were deeply anesthetized and perfused with 1× phosphate-buffered saline (PBS) and ice-cold 4% paraformaldehyde (PFA) in PBS, pH 7.2. Brains were dissected and postfixed with 4% PFA overnight at 4°C. For cryoprotection, fixed brains were stored in 30% sucrose in PBS at 4°C. The brain was embedded in optimal cutting temperature compound (Tissue Tek, Sakura Finetek, 25608-930) and frozen at −80°C for cryosectioning. Frozen brains were serially sectioned with Leica CM 1850 or 1950 (Leica Microsystems, Wetzlar, Germany) in the coronal plane at 16 µm thickness. Every 15th sections (each slice 240 µm apart from the next) were serially mounted on individual Colorfrost Plus Microscope Slides (Fisher Scientific) in order from anterior to posterior and preserved at −20°C until use.

## LacZ staining

Animals for LacZ staining were perfused with PBS, and the dissected brains were postfixed with 2% PFA for 1.5 hr at 4°C. Cryosections were washed with PBS, and X-gal staining was developed at 37°C overnight in the staining solution (5 mM $K_3Fe(CN)_6$, 5 mM $K_4Fe(CN)_6$, 5 mM EGTA(ethylene glycol-bis(β-aminoethyl ether)-N,N,N′,N′-tetraacetic acid), 0.01% deoxycholate, 0.02% NP40, 2 mM $MgC1_2$, and 1 mg/ml X-gal). Sections were postfixed with 10% formalin at room temperature overnight, followed by counterstain with nuclear-fast red (H-3403, Vector Laboratories) at room temperature for 10 min before proceeding for dehydration (70%, 95%, 100% ethanol, xylene twice) and coverslipping with Mount-Quick (Ted Pella).

## Immunohistochemistry

Cryosections were washed with PBS and incubated overnight at 4°C with primary antibodies diluted in blocking solution (10% Lamb serum and 0.3% Triton X-100). The following primary antibodies were used in this study: rabbit anti-Sox2 (1:1000; Abcam, ab92494, RRID:AB_10585428); rat anti-Sox2 (1:1000; Invitrogen, 14-9811-82, RRID:AB_11219471); rabbit anti-DCX (1:1000; Abcam, ab18723, RRID:AB_732011); chicken anti-GFAP (1:1000; Millipore, AB5541, RRID:AB_177521); rat anti-RFP (1:1000, Chromotek, 5f8-100, RRID:AB_2336064); chicken anti-S100B (1:500, Synapyic Systems, 287 006, RRID:AB_2713986); rat anti-BrdU (1:500, Abcam, ab6326, RRID:AB_305426) and mouse anti-BrdU (1:100, BD Biosciences, 347580, RRID:AB_400326); rabbit anti-GluR2/3 (1:100, Millipore, AB1506, RRID:AB_177521), rabbit anti-GluR2/3 (1:500, Epitomics, 1905), mouse anti-c-fos (1:500; Novus Biologicals, NBP2-50037, AB_2665387), rabbit anti-c-fos (1:1000, Synapyic Systems, 226 003, AB_2231974), rabbit anti-c-fos (1:100, Synapyic Systems, 226 008, RRID:AB_2891278). For staining of Sox2 and thymidine analogs, sections were heated in 10 mM citric acid pH 6.0 on boiling water bath for 15 min prior to blocking. After three washes in PBS, sections were incubated for 2 hr with corresponding secondary antibodies; goat anti-rat IgG (H+L) Alexa Fluor 546 (Invitrogen, A-11081, RRID:AB_2534125), donkey anti-rabbit IgG (H+L) Alexa Fluor 546 (Invitrogen, A-10040, RRID:AB_2534016), goat anti-chicken IgY (H+L) Alexa Fluor 546 (Invitrogen, A-11040, RRID:AB_2534097), goat anti-rabbit IgG (H+L) Alexa Fluor 633 (Invitrogen, A-21070; RRID: AB_2535731), donkey anti-mouse IgG (H+L) Alexa Fluor 647 (Invitrogen, A-31571, RRID:AB_162542), donkey anti-rabbit IgG (H+L) Alexa Fluor 488 (Invitrogen, A-21206,RRID:AB_2535792), and goat anti-rabbit IgG (H+L) Alexa Fluor 488 (Invitrogen, A-11008,RRID:AB_143165). Nucleus were stained by 4′,6-diamidino-2-phenylindole, dihydrochloride (Thermo Fisher, D1306). After a final rinse with PBS, sections were mounted on glass slides with Prolong gold antifade reagent (Thermo Fisher Scientific, P36930).

## Single mRNA detection by RNAscope

RNAscope in situ hybridization combined with immunofluorescence were performed using the RNAscope Multiplex Fluorescent V2 Assay (Advanced Cell Diagnostics) according to manufacturer's protocol. Briefly, sections were dehydrated by ethanol and then treated with preheated co-detection target retrieval reagent (Advanced Cell Diagnostics). Subsequently, sections were incubated with primary antibody for rabbit anti-GluR2/3 (1:100, Millipore, AB1506, RRID:AB_177521) overnight

at 4°C and post-fixed with 10% neutral buffered formalin. After three washes, the sections were treated with Protease Plus (Advanced Cell Diagnostics), and incubated with RNAscope probe for *Shh*, designed commercially by Advanced Cell Diagnostics (314361, RNAscope LS 2.5 Probe-Mm-Shh) for 2 hr at 40°C in HybEZ Oven (Advanced Cell Diagnostics). Detection of probe and antibody were performed using Tyramide Signal Amplification system with Opal 570 or 520 reagents (Akoya Biosciences, FP1488001KT and FP1487001KT). Expression of *Shh* in each mossy cell was evaluated by the number of Shh probe puncta in a GluR2/3-positive cell body. *Shh* mRNA signals in GluR2/3+ cells were counted in a series of sections spanning from anterior to posterior DG per mouse.

## Cell counting

Images were acquired using a CSU-W1 spinning disk confocal microscope (Nikon) and Axioscan Z.1 (Carl Zeiss). NIH ImageJ (RRID:SCR_003070) was used for cell counting. Cell counting was performed in the DG of the hippocampus in a series of sections that were collected by serial mounting on 15 individual glass slides. Marker-positive cells in the indicated areas of the DG were counted in a series of sections on a single glass slide that contained DG at the same anatomical level between each group. The counts were then multiplied by the number of intervals between sections to estimate the total number of marker-positive cells in the DG. To quantify c-fos+ cell density in dentate hilus, c-fos+ cells were manually counted and normalized to the area size of hilus that are measured by ImageJ. For counting the total number of NSCs in the DG, radial NSCs were identified with a radial GFAP+ process extending from a Sox2+ nucleus in the subgranular zone of the DG. Persistence of NSC identity in each animal was calculated by dividing the total number of BrdU+ Sox2+ GFAP+ radial NSCs at 10 days or 4 weeks after BrdU administration by the total number of BrdU+ Sox2+ NSCs on the day of last BrdU administration (day 0) in the same group.

## Statistical analysis

Statistical analyses were performed using either two-tailed unpaired *t*-test (for comparisons between two groups); one- or two-way analysis of variance with Tukey's multiple comparison test (for multiple groups comparison) with Prism 9.0 software (GraphPad, RRID:SCR_002798). Differences were considered statistically significant at $p < 0.05$. Asterisks indicate significant differences (*<0.05; **<0.01, ***<0.001, ****<0.0001). No statistical methods were used to predetermine sample sizes. Sample sizes were based on previous studies using similar methods (*Harris et al., 2021*; *Jessberger et al., 2007*; *Parent et al., 1997*; *Sierra et al., 2015*; *Yeh et al., 2018*). Statistical analysis for specific experiments is shown in figure legends.

## Acknowledgements

We are very grateful to the members of the S.J.P. lab for helpful discussions, in particular Dr. M Lun for technical help, suggestions, and helping to write this manuscript and T Huynh and J G Castillo for technical help. We thank Dr. Zachary Knight (University of California, San Francisco) for DREADD mice. This research was supported by NIH grant R01 NS118995 (S.J.P.), The Uehara Memorial Foundation (H.N.), JSPS Overseas Research Fellowships (H.N.). We also thank the Nikon Imaging Center at UCSF. Confocal microscopy with the CSU-W1 spinning disk was supported by the S10 Shared Instrumentation grant (1S10OD017993-01A1).

## Additional information

### Competing interests

Samuel Pleasure: Reviewing editor, eLife. The other authors declare that no competing interests exist.

### Funding

| Funder | Grant reference number | Author |
| --- | --- | --- |
| National Institutes of Health | R01NS118995 | Samuel Pleasure |

| Funder | Grant reference number | Author |
|---|---|---|

The funders had no role in study design, data collection, and interpretation, or the decision to submit the work for publication.

## Author contributions

Hirofumi Noguchi, Conceptualization, Data curation, Formal analysis, Validation, Investigation, Methodology, Writing - original draft; Jessica Chelsea Arela, Thomas Ngo, Data curation, Investigation; Laura Cocas, Data curation, Investigation, Methodology, Writing – review and editing; Samuel Pleasure, Conceptualization, Resources, Supervision, Funding acquisition, Visualization, Methodology, Project administration, Writing – review and editing

## Author ORCIDs

Hirofumi Noguchi ⓘ http://orcid.org/0000-0002-9779-4956
Samuel Pleasure ⓘ https://orcid.org/0000-0001-8599-1613

## Ethics

All animal experimentation in this study were approved by the IACUC committee at UCSF (protocols AN191848 and AN200243) and were performed in strict accordance with the recommendations in the Guide for the Care and Use of Laboratory Animals of the National Institutes of Health.

Reviewer #1 (Public Review): https://doi.org/10.7554/eLife.91263.2.sa1
Reviewer #2 (Public Review): https://doi.org/10.7554/eLife.91263.2.sa2
Author Response https://doi.org/10.7554/eLife.91263.2.sa3

# Additional files

## Supplementary files

• MDAR checklist

## Data availability

All data generated or analyses during this study are included in the manuscript or in uploaded excel data files.

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
