## [Editor Report · eLife assessment]

This study uses specific and robust genetic approaches to assess mechanisms of kainic acid-induced neurogenesis. This is a **fundamental** study that bridges several complementary methods and is a **convincing** use of existing approaches to explore roles for sonic hedgehog in activity-dependent and aging-associated hippocampal neurogenesis.

---

## [Referee Report · Reviewer #1 (Public Review)]

Summary: Seizure stimuli has long been recognized to exhibit potent effects on adult neurogenesis, from depletion of the NSC pool to promoting aberrant migration of adult-born neurons. However, the identity and source of extrinsic signals is still incompletely understood. The work by Noguchi et al., demonstrates that Shh from mossy cells is a major source of Shh signaling after KA-mediated acute seizures. This work is interesting because mossy cells undergo hyperactivation during seizures, so this study provides a mechanistic link between mossy cell neuronal activity control of neurogenesis through Shh signaling. Weaknesses are that only male mice were analyzed in the seizure induction experiments and several control groups are missing for seizure induction, tamoxifen induction, and the DREADD experiment.

Strengths:

1. The study uses rigorous and specific genetic approaches (e.g., GliLacZ/+ mice; ShhEGFP-Cre/+ mice; mossy cell selective conditional Shh knockout using Crlr-Cre mice) to demonstrate Shh signaling is activated by seizures in mossy cells and contributes to aberrant neurogenesis.

2. Use of DREADDs (Crlr-Cre; hM3Dq) to show mossy cells control adult neurogenesis through Shh in an activity-dependent manner.

3. Demonstration that Shh deletion in mossy cells leads to reduction of the NSC pool uses stringent methods and analysis, including BrdU pulse-chase and co-labeling with NSC markers.

Weaknesses:

1. The analysis of Shh deletion in mossy cells and influences of aging related NSC pool decline is not well connected with the rest of the study on the expression/requirement of Shh in mossy cells to regulate seizure-induced neurogenesis. To promote cohesion, the authors should examine/discuss what happens to mossy cells during aging - it is similar or different to what happens to mossy cell neuronal activity during seizures?

2. Only male mice were analyzed in the seizure induction experiments, leaving open the possibility of sex differences since previous reports suggest sex differences in adult neurogenesis.

3. Several control groups are missing:

-For seizure induction: missing vehicle (instead of no KA treatment).

-For TAM induction: missing corn oil only to check leakiness and specificity of transgene.

-For DREADD experiment: missing vehicle (to control for hM3 non-specific effects)

---

## [Referee Report · Reviewer #2 (Public Review)]

Summary:

The mechanisms by which seizures induce neurogenesis has remained unclear. Prior work from the authors demonstrated Mossy cell expressed Shh, that altered Shh expression follows epileptic seizures, and that Shh is a neural mitogen. Here authors show that Shh from mossy cells, which are well positioned between the pyramidal and granule cell layers, are a major source of signaling after seizures, contributing to seizure-induced neurogenesis. Moreover, they find that Mossy cell-sourced Shh is required for self-renewal of NSCs even outside of the context of seizures.

SVZ Gli1 expression was detected in NSCs and Gli1 reporter activity follows kainate-induced seizures. Heterozygous Shh mice show reduced seizure induced Shh signaling and reduced neurogenesis. After localizing Shh production to Mossy cells, authors removed Shh from Mossy cells and found reduced neurogenesis. By activating mossy cells through chemogenetic DREADD, they found that the effect of mossy cells on SVZ neurogenesis is activity-dependent, that Shh signaling activity is upregulated in NSCs by mossy cell neuronal activity, and that the induction of neurogenesis by mossy cell neuronal activity is compromised in the absence of Shh from mossy cells. In a series of experiments incorporating AAV DREADD, they find that mossy cell activity can contribute to neurogenesis in contralateral DG, and that seizure induced Shh may be transported along mossy axons. To examine long-term effects, they study mice several weeks after seizure, and find that suggesting that NSCs are less likely to return to their stem cell state after seizure-induced proliferation in the absence of Shh from mossy cells, and that Shh from mossy cells contributes to persistence of the NSC state during aging.

Strengths:

The results are compelling and impactful, and the study is extremely well done. The various genetic lines in the study ensure robust results. Adequate consideration of statistics, methods of quantification, and avoidance of artifact is given.

Weaknesses:

None identified.

---

## [Author Response]

1. The analysis of Shh deletion in mossy cells and influences of aging related NSC pool decline is not well connected with the rest of the study on the expression/requirement of Shh in mossy cells to regulate seizure-induced neurogenesis. To promote cohesion, the authors should examine/discuss what happens to mossy cells during aging - it is similar or different to what happens to mossy cell neuronal activity during seizures?

We believe that both are similar mechanisms. Seizure induced neurogenesis increases NSC proliferation, which increases demand of Shh to increase self-renewal. Similarly, we assume that increased NSC decline in Shh cKO mice is due to the increased demand of Shh for self-renewal of NSC with aging. It has been shown that NSCs in young mice generally don’t self-renew and instead are consumed after one or two rounds of cell division. On the other hand, NSCs in old mice are known to undergo more rounds of cell division compared with younger mice. This suggests that NSCs may be more dependent on signals driving self-renewal in aged-mice. Our suggestion is that Shh from mossy cells contributes to minimising the NSC pool decline with aging, and therefore loss of Shh from mossy cells results in increased decline of the NSC pool in aged-Shh cKO mice. This aligns with our hypothesis that Shh from mossy cells contributes to maintenance of the NSC pool.

What is the exact mechanism regulating the shift of proliferation capacity of NSC with aging remains unclear and would be an interesting topic for future studies. In addition, whether mossy cell neuronal activity is decreased with age or Shh release/expression is compromised in aged animals remains to be elucidated. Considering these factors together, the brain region(s) and other factors that regulate neuronal activity of mossy cell thereby controlling Shh release and how these are dysregulated in pathological conditions and in aging will be important studies for future research.

1. Only male mice were analyzed in the seizure induction experiments, leaving open the possibility of sex differences since previous reports suggest sex differences in adult neurogenesis.

Seizure induced neurogenesis was observed in both male and female mice. Considering that, we assumed that mossy cell derived Shh regulates seizure induced neurogenesis also in female mice. However, we agree with the reviewers’ comments. We can not exclude the possibility that female mice reacts to KA or seizures differently from male mice, or that Shh from mossy cells might have distinct effects in female mice in that paradigm. It is also an interesting possibility that female specific behaviors may affect mossy cell activation and also regulate neurogenesis though Shh. Because these are large and unresolved questions, we elected to leave potential sex difference in mossy cell regulated neurogenesis for future research.

1. Several control groups are missing:-For seizure induction: missing vehicle (instead of no KA treatment).-For TAM induction: missing corn oil only to check leakiness and specificity of transgene.-For DREADD experiment: missing vehicle (to control for hM3 non-specific effects)

About missing vehicles in KA treatments, we used saline (0.9% NaCl) as a vehicle for Kainic acid, which is commonly used as a vehicle for water soluable reagents in adult neurogenesis experiments. In addition, the average volume of KA solution that mice received intrapenitorially for seizure induction was less than 500ul, which is less than recommended maximum volume in NIH and UCSF. We have not tested if the saline injection makes a difference in our experiments but based on previous reports using saline, we believe that saline would not affect our experimental results.

About Tamoxifen injections, the Gli1-CreER mice have been widely used for fate tracing analysis including in our previous research where Gli1-CreER mice have shown specific recombination in Gli1-expressing NSCs. Our results in this study have shown consistently that Gli1-CreER;;Ai14 mice label NSCs in the dentate gyrus. Given this, we believe that our result using Gli1-CreER line are not affected by non-specific recombination without tamoxifen.

About Clozapine (CZL) injection, we decided to administer CLZ in both control and DREADD animals considering the possible side-effects of CLZ. We agree with the reviewer that our experiment cannot exclude the possibility that expression of hM3Dq affects neurogenesis without CLZ or CNO. However, although we have not included the analysis using saline as a control in our experiments, we have tested that both transgenic and virus-injected mice DREADD expressing mice respond to CLZ and activate neuronal activity of mossy cells compared with control animals. Therefore, we believe that it does not affect the interpretation of our data that mossy cell neuronal activity controls neurogenesis.

We appreciate reviewers' carefully considered comments and we will apply suggested controls to our future research.